**Data Availability Statement:** Data are available at https://osf.io/yz6h2/.

**Funding:** Funding for this research was provided by the National Science Foundation (https://www.nsf.gov/) via grants NSF CAREER BCS-0748717 (to

# Are there dedicated neural mechanisms for imitation? A study of grist and mills

**Elizabeth Renner**[1,2]*, **Yishan Xie**[3], **Francys Subiaul**[1,4], **Antonia F. de C. Hamilton**[3,5]

**1** Center for the Advanced Study of Human Paleobiology, Department of Anthropology, The George Washington University, Washington, DC, United States of America, **2** Department of Psychology, Northumbria University, Newcastle-upon-Tyne, United Kingdom, **3** Institute of Cognitive Neuroscience, University College London, London, United Kingdom, **4** Department of Speech, Language, and Hearing Sciences, The George Washington University, Washington, DC, United States of America, **5** Department of Psychology, University of Nottingham, Nottingham, United Kingdom

* lrenner@gwmail.gwu.edu

## Abstract

Are there brain regions that are specialized for the execution of imitative actions? We compared two hypotheses of imitation: the mirror neuron system (MNS) hypothesis predicts frontal and parietal engagement which is specific to imitation, while the Grist-Mills hypothesis predicts no difference in brain activation between imitative and matched non-imitative actions. Our delayed imitation fMRI paradigm included two tasks, one where correct performance was defined by a spatial rule and another where it was defined by an item-based rule. For each task, participants could learn a sequence from a video of a human hand performing the task, from a matched "Ghost" condition, or from text instructions. When participants executed actions after seeing the Hand demonstration (compared to Ghost and Text demonstrations), no activation differences occurred in frontal or parietal regions; rather, activation was localized primarily to occipital cortex. This adds to a growing body of evidence which indicates that imitation-specific responses during action execution do not occur in canonical mirror regions, contradicting the mirror neuron system hypothesis. However, activation differences did occur between action execution in the Hand and Ghost conditions outside MNS regions, which runs counter to the Grist-Mills hypothesis. We conclude that researchers should look beyond these hypotheses as well as classical MNS regions to describe the ways in which imitative actions are implemented by the brain.

## Introduction

Humans are prolific imitators and make extensive use of this ability to learn from and connect with others [1]. Imitation has been localized to the premotor and parietal cortex in a large number of studies [2–7]. These regions have been widely believed to contain mirror neurons [8].

One theory of how the brain produces imitative actions, which we call the mirror neuron system (MNS) hypothesis, suggests that imitation engages dedicated brain mechanisms [9–12]. Mirror neurons respond both when an individual sees another carry out an action and when that individual performs the same action. The MNS hypothesis holds that the superior temporal

F.S.) and NSF-IGERT DGE-080163; the Leakey Foundation (to E.R., https://leakeyfoundation.org/); and the European Research Council (https://erc.europa.eu/homepage) via consolidator grant INTERACT 313398 (to A.F.D.C.H.). The funders had no role in study design, data collection and analysis, decision to publish, or preparation of the manuscript.

**Competing interests:** The authors have declared that no competing interests exist.

sulcus (STS) works with the frontoparietal mirror neuron system—inferior frontal gyrus, adjacent premotor cortex, and inferior parietal lobule—to process the visual, motor, and goal components of imitative actions [9, 13]. Specifically, the STS processes the higher-order visual aspects of an observed action; it feeds this information to the frontoparietal mirror neuron system (inferior frontal gyrus and inferior parietal lobule), which processes the goal of the action and its motor specification; this information is then fed back to the STS, which matches predicted sensory consequences and the visual aspects of the planned action [9].

Imitation could be a unique skill subserved by a dedicated brain system (as in the MNS hypothesis) or may be the product of more general sensorimotor processing. A recent prominent theory suggests the latter. Heyes' "cognitive gadgets" theory [14–16] uses an analogy of grist and mills to explain neurocognitive imitation mechanisms. According to this view, general sensorimotor mechanisms (mills) can receive as input either social cues (one type of grist) or non-social cues (another type of grist), processing both in the same way. When the input signal is social, the output might be labelled as imitation, but Heyes argues that both social and non-social signals are *processed* in essentially the same way. In this model, excitatory vertical associations form between sensory stimuli and their corresponding motor actions. The sensory stimuli "can be in any sensory modality and originate from animate or inanimate objects" ([16], p. 126). Furthermore, the "degree of topographic resemblance between observed and executed actions" is never calculated ([16], p. 127; that is, there is no topographic matching process, as the excitatory connections are sufficient to enable imitation. We will refer to this view as the Grist-Mills hypothesis. Surprisingly, no studies that we are aware of have directly contrasted these two hypotheses in the case of motor outputs.

Several brain imaging studies have compared conditions where participants imitate to those where they do not, and often report more engagement of mirror neuron system (MNS) regions during imitation. In most of these studies, the visual input and corresponding motor response are present simultaneously. For example, Iacoboni and colleagues [17] asked participants to lift their middle or index finger in response to either dynamic social stimuli (a video of a hand lifting a finger), static social stimuli (an image of a hand with the finger to be lifted marked by an "x"), or non-social stimuli (an "x" displayed in a certain spatial location on a gray background). Participants initiated their finger motions concurrently with the displayed stimuli. These conditions were compared to visually identical conditions in which participants observed the same stimuli but did not perform any actions. Frontal and parietal regions were found to be more active in the imitation condition relative to controls. However, in this paradigm (and many others), the visual stimulus differs between the imitation condition (video) and the non-imitation conditions (static image and symbolic instruction), so it is not possible to know whether the observed differences in brain activity are due specifically to the imitation of actions or to differences in input more generally.

To address this problem, several neuroimaging studies have examined delayed (as opposed to concurrent) imitation. Buccino and colleagues [2] showed participants video clips of hands forming guitar chords, which were imitated after a pause; however, the condition in which observed actions were imitated was not directly contrasted with one in which chords were freely executed. Similarly, participants in a study by Krüger and colleagues [6] saw video clips of hands making extension-flexion movements at the wrist. These were also imitated either immediately or after a pause, but there was no equivalent condition in which motions were instructed in a non-imitative context. Chaminade, Meltzoff, and Decety [18] compared the imitation of actions on Lego blocks to freely chosen actions on the same sets of blocks, with a brief delay between observation and response; however, it is unclear if these actions are matched for motor components. Makuuchi [19] examined immediate and delayed imitation versus symbolically instructed finger and wrist actions; only visual area V5 was more activated

for imitation than symbolic instruction, and it was concluded that Broca's area/inferior frontal cortex is not required for imitation. While some of the above studies report activation in parts of the MNS (e.g., inferior parietal cortex [2, 18]), delayed imitation conditions were not necessarily contrasted with matched control conditions. The results of these studies do not show clear involvement of the MNS in the execution phase of delayed imitation.

Here, we test the MNS and Grist-Mills hypotheses using a dataset collected in the study of different types of delayed imitation [20]. Our original design was adapted from previous behavioral imitation studies with children [21, 22], monkeys [23, 24], and orangutans [25]. These are delayed imitation tasks in which participants first see a sequence of 3 or more actions, and after a short delay they perform the same sequence. Two different types of sequence are commonly used: in the "cognitive task", the correct responses are defined according to the identity of the objects selected (for example, pear → pliers → basketball; Fig 1A, Ghost example); in the "spatial task", the correct responses are defined according to the locations selected (for example, bottom → top-right → center-left; Fig 1B, Ghost example).

A key feature of these delayed imitation tasks is that the correct sequence can be learned in many different ways, for example by watching another person perform the sequence (imitation), by reading a list of instructions, or by trial and error. Here, we contrast imitation, where a human hand demonstrates the correct sequence in a video, to two alternative (non-imitative) learning conditions. Trial-and-error learning was not used.

Different types of learning occur during the demonstration phase of the task, while the execution phase is the same across conditions. In Hand demonstrations, participants saw at the center of the screen a video of a human hand moving a joystick to select picture items (Fig 1A and 1B, Hand examples); when selected, a blue frame appeared around the picture and then the picture vanished. Thus, participants could learn the observed sequences by copying the demonstrated hand-joystick actions along with their effects (i.e., imitation learning). Ghost demonstrations were identical to Hand demonstrations except that the middle of the screen remained blank. Hence, participants could learn the sequence vicariously, using the blue frames and pictures' subsequent disappearance as indexes of serial order without any observed hand-joystick actions (Fig 1A and 1B, Ghost examples). In Text demonstrations, the correct sequence was shown in words which vanished one at a time with exactly the same timing as did images in Hand and Ghost demonstrations (Fig 1A and 1B, Text examples). The Text condition was included to control for covert speech when learning the sequence order. All three demonstration conditions imposed the same demands on working and serial memory; the only difference was the format in which the sequence information was presented [20].

Each trial's demonstration phase was followed by an execution phase, in which participants performed the sequence they observed. On the response screen for the cognitive task, the same images from the demonstration phase were shuffled into a different spatial configuration (Fig 1A, Hand, shows an example of what a participant might see during a demonstration phase, and Fig 1C shows what they would see during the corresponding execution phase). On the response screen for the spatial task (Fig 1D), a different set of three identical pictures was displayed in the same spatial locations as in the demonstration phase. Participants executed the previously demonstrated cognitive or spatial sequence by selecting images in the target order using a joystick.

This study was originally implemented to examine brain regions involved in sequence imitation, and we previously reported results from the demonstration phase [20]. However, this design provides a unique opportunity to examine Heyes' Grist-Mills hypothesis [14] and the MNS hypothesis, with two notable strengths. Firstly, there is a clear temporal distinction between the demonstration phase (observation) and the execution phase (response). After each trial's demonstration phase is an interstimulus interval (black screen) with a duration of 1

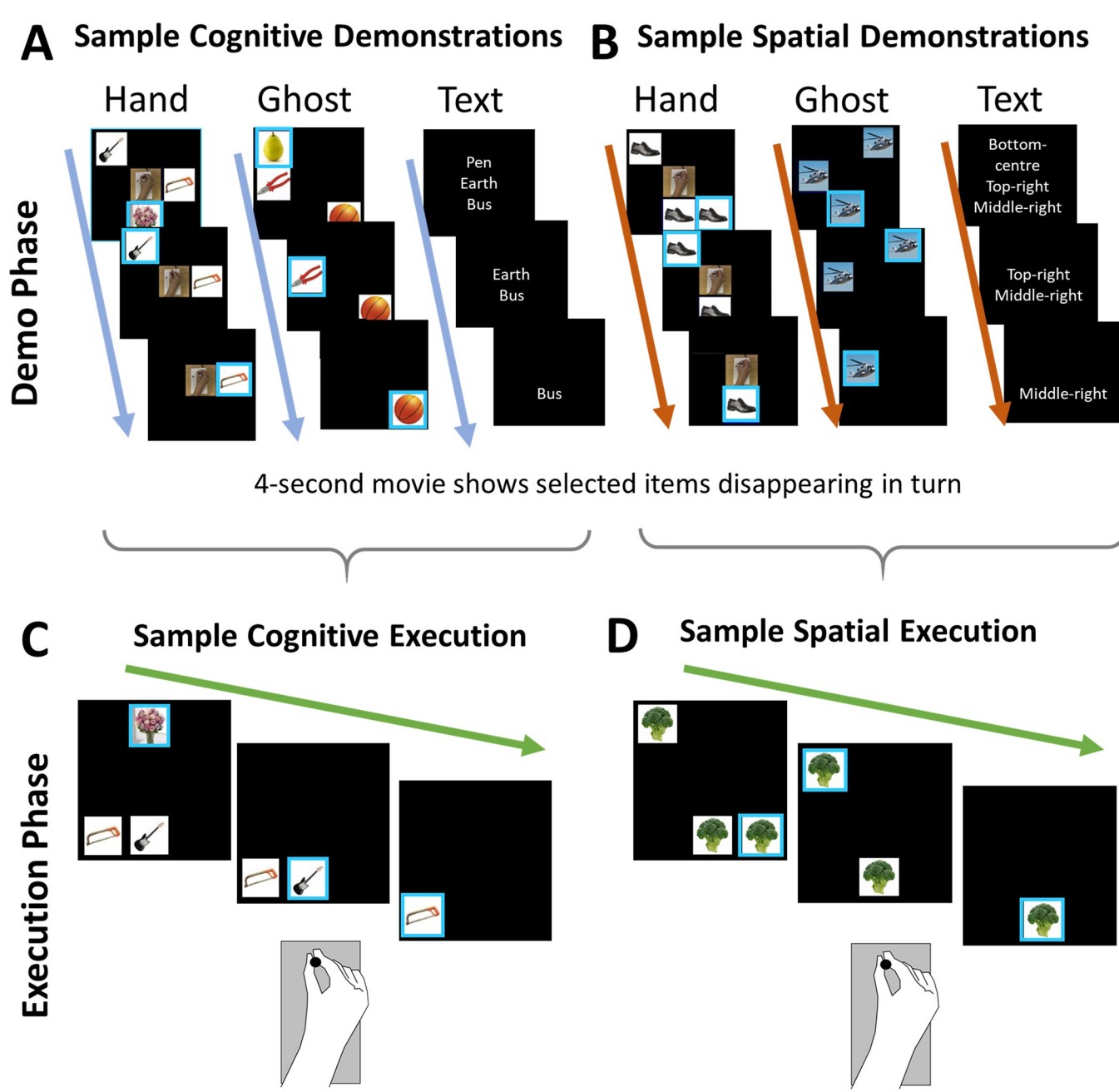

**Fig 1.** Examples of demonstration phases (A and B) and execution phases (C and D) from the cognitive and spatial tasks. For both tasks, there are three possible ways to learn the sequence (Hand, Ghost, and Text). In the Hand condition, a small video in the center of the display shows the hand of a person performing the sequence; in the Ghost condition, the sequence is indicated by the movement of the blue square frame and disappearance of each picture; and in the Text condition, the rules are shown in English words which disappear. There is only one possible way to execute the sequence—by selecting each item using the joystick (C and D). For both cognitive and spatial sequences, only the execution trials which follow a Hand demonstration are classed as imitation. However, only in the spatial task could participants' joystick responses correspond with those observed, as pictures' spatial arrangement was constant. In the cognitive task, because picture location varied randomly, so did joystick responses. (C) Example of the display during the execution phase of the cognitive task (i.e., all response screens after any demonstration type in the cognitive task could look like this, but with image content determined by the preceding demonstration phase). The three items appear in different locations and the participant must use the joystick to select each in turn so they disappear. (D) Example of the display during the execution phase of the spatial task.

to 4 seconds. Thus, overt visual confounds are controlled during the execution phase (as in [26]), while overt motor confounds are controlled during the demonstration phase. Secondly, in the execution phase, the visual stimuli are carefully controlled. The visual stimuli are matched when participants execute sequences across all demonstration conditions, as shown in Fig 1C and 1D. Thus, any difference in brain activation during action execution following different demonstration conditions cannot be explained by varying visual input during the execution phase—as there was none; rather, it must be attributed to different underlying mechanisms for learning and implementing the action sequence.

Here we present a detailed analysis of the execution phase of this task in order to test competing hypotheses for the mechanisms underlying imitation. For instance, the MNS hypothesis proposes that the MNS forms a core or specialized neural circuitry for imitation, and that observed and performed actions are matched by mirror neurons [9, 11]. Because mirror neurons are active both when a particular action is observed and when it is performed, we should see activation in MNS regions when participants execute an imitation response (Hand-cognitive and Hand-spatial conditions) compared to other non-imitative conditions (Ghost-cognitive and Ghost-spatial conditions). The Grist-Mills hypothesis [14] predicts that regions implementing imitative responses are identical to those that implement non-imitative responses. Consequently, it predicts no differences between the execution phases of the Hand-cognitive and Hand-spatial conditions compared to the matched non-social (Ghost and Text) control conditions.

## Materials and methods

We previously reported an fMRI study of imitation learning where every trial comprised a demonstration phase in which participants learned a sequence and an execution phase in which they executed it [20]. The previous paper analyzed the demonstration phase, and here we analyze the execution phase. As the details of the implementation of the study and the trials are given in full elsewhere [20], we provide only an essential outline here.

### Participants

The protocols of this study were approved by the University of Nottingham Ethics Committee. A total of 23 adults were recruited from September to November 2012 and gave written informed consent to take part in the study. Researchers had access to personally identifiable participant data during and after data collection. All participants were right-handed, and had normal or corrected-to-normal vision and no history of neurological disorders. Data from four participants who made excessive errors (>30% of trials) or completed only one scanning session were excluded. Nineteen participants were included in the analysis (10 female; median age = 20).

### Experimental design

This is an event-related fMRI study with a 2 × 3 factorial design, with two task types (cognitive and spatial) and three demonstration types (Hand, Ghost, and Text). In the cognitive task, images of three different items appeared on the screen simultaneously. Participants were required to select images in an order based on the identity of the items, regardless of their positions. The spatial arrangement of the images varied randomly between the demonstration and execution phases. In order to correctly perform the task, participants had to construct an abstract "cognitive" representation of the identity of the images, rather than a motor or spatial representation. For example, the correct sequence of Fig 1A (Hand) and 1C is flowers → guitar → saw.

In the spatial task, three identical images appeared on the screen, and the spatial arrangement of the items was the critical factor. Participants were required to select the pictures in an order based on their spatial locations, regardless of the content of the pictures. The contents of the pictures changed between the demonstration and execution phases. The sequence shown in Fig 1B (Ghost) is selected in the order bottom-center → top-right → middle-left.

The two tasks were each performed with three different types of demonstration. In a Hand demonstration, a video of a human right hand in first-person perspective operating an fMRI-compatible joystick was presented at the centre of the screen (Fig 1A and 1B, Hand conditions). As the hand moved the joystick toward a particular location, a blue square frame appeared around the image to indicate that the corresponding image was selected. The image then disappeared, signaling that it had been correctly chosen. Thus, in this condition participants learned the sequence rules by observing and copying the actions of another human agent; this condition represents imitation learning.

In a Ghost demonstration, participants saw the same type of sequence of item selection, but the center of the screen was left blank (Fig 1A and 1B, Ghost conditions). As in the Hand demonstrations, a blue square appeared around each image in turn to indicate that the corresponding image was selected; the image then disappeared. Hence, participants learned the sequence rules vicariously, via physical cues (appearance of frame followed by disappearance of the image) rather than social cues (hand movements) provided by an agent, making this condition equivalent to non-social learning.

In a Text demonstration, the correct sequence was shown in English words which vanished one at a time with the same timing as the images disappearing in Hand and Ghost demonstrations (Fig 1A and 1B, Text conditions). Therefore, participants learned the sequence rule by reading text instructions. This condition was included to control for covert speech when learning the sequence order.

In each execution phase, participants performed the sequence rule from the preceding demonstration phase. In this phase, the visual stimuli were carefully controlled, as shown in Fig 1C (this execution phase could have been preceded by any type of demonstration in the cognitive task) and D (which could have been preceded by any type of demonstration in the spatial task). All responses were made using an MRI-compatible joystick; as the participant operated the joystick, a blue square frame appeared around the selected picture, and the corresponding picture vanished, indicating a correct response. This procedure paralleled those witnessed in the Hand and Ghost demonstrations. Custom-written Cogent scripts in Matlab were used to run the experiment.

Participants received feedback on their performance on each trial. If a picture was selected correctly, it disappeared. Therefore, selecting three pictures in a correct sequence led to the vanishing of all pictures. Two types of error could occur: incorrect selection and slow response. When an image was chosen in the incorrect order in a sequence, the word "Error" appeared on the screen until the beginning of the next trial. Also, if the correct response was not made within the allotted time (6 seconds), the word "Error" was shown briefly on the screen before the beginning of the next trial. Participants were instructed to respond as accurately and quickly as possible.

The timings were as follows. The demonstration phase duration was 4 seconds. An inter-stimulus interval of 1 to 4 seconds was imposed after the demonstration phase. The execution phase duration was 6 seconds. An intertrial interval of 3 or 7 seconds was imposed after the execution phase of one trial and before the demonstration phase of the next trial.

The present study focuses only on the execution phase, in which the visual and motor elements of the task were matched across all 6 conditions. Only the manner in which the participants learnt the sequence in the previous (demonstration) phase differed.

## fMRI scanning

Participants became acquainted with the tasks by performing 48 practice trials in the fMRI scanner; during this time, T1 anatomical scans were recorded. Subsequently, two sessions of 48 trials were carried out. Participants took a break between the two sessions. Trials were drawn from the six cells of $2 \times 3$ factorial design, with 8 trials in each cell. Each session's trial order was pseudorandomised by permutation of the complete set of possible trials.

fMRI scanning was carried out at the University of Nottingham, in a 3T Phillips scanner (Philips Medical Systems, Best, The Netherlands) with the following settings: double echo imaging, 37 slices per TR (thickness = 3 mm), TR = 2500 ms, TE = 20 and 45 ms, flip angle = 80˚, matrix = $64 \times 64$, field of view = 19.2 cm. 308 images were collected for each session. Double echo imaging was adopted to enhance signal detection [27]. Images of the two sessions were combined by weighted summation according to the signal strength in each brain area [28].

## Data analysis

Data and statistical analyses were conducted in the Statistical Parametric Mapping package (SPM12; The Wellcome Centre for Human Neuroimaging, London, UK) implemented in Matlab 2018a (Mathworks, Natick, MA). Preprocessing comprised realignment and unwarping, normalization, and smoothing. For each participant, images were realigned to the first image of session 1 to resolve head movement. The mean image of the 308 realigned images was adopted as an individual parameter for normalization into the standard EPI template. The normalized images were then smoothed with an 8 mm × 8 mm × 8 mm Gaussian filter to resolve neuroanatomical variability between subjects and to increase signal-to-noise ratio. After preprocessing, a registration check was conducted for excessive head movement and to ensure that the preprocessed brains were aligned correctly. Default SPM parameters were used, unless specified otherwise. All 19 participants passed the brain movement check. The registration check showed that the very top part of the brain of one participant had not been recorded; this did not impact on the MNS regions that we focus on, so data from all 19 participants were included.

For first-level analysis, a design matrix modeling demonstration and execution events according to their category was fitted for every participant. Demonstration and execution events have 4-second and 6-second durations, respectively. When errors occurred, both the demonstration and execution event of the trial were modelled in separate "demo-error" and "exe-error" categories. Therefore, there were 14 regressors for each session.

## Statistical analysis

Second-level contrasts were computed for the demonstration and execution phases separately; this paper addresses only the execution phase. We focus on the contrast between execution following a Hand demonstration and execution following a Ghost demonstration, because this is most relevant to our hypothesis. Thus, we evaluate and report here the simple effects of Hand-cognitive > Ghost-cognitive and of Hand-spatial > Ghost-spatial. Other simple effects are listed in tables in S1 File. Statistical thresholds were first set to voxelwise $p < 0.0001$ uncorrected, and we report only clusters that survive a $p < 0.05$ FWE corrected. The cluster-forming extent threshold $k$ was 10.

Clusters were assigned anatomical labels using the xjView toolbox (https://www.alivelearn.net/xjview).

## Results

### Cognitive task

Executing a cognitive task sequence after observing a Hand demonstration in contrast to a Ghost demonstration resulted in greater activation in multiple cortical areas, including right calcarine cortex (Fig 2B), left middle occipital region (Fig 2A), bilateral precuneus, right parahippocampus, and left amygdala and hippocampus (Table 1). There was no activity in premotor or parietal cortex for this contrast, even at a lower threshold.

### Spatial task

Executing a spatial task sequence after observing a Hand demonstration in contrast to a Ghost demonstration resulted in greater activation in right lingual gyrus (Fig 2C) and left cerebellum (Table 2). There was no activity in premotor or parietal cortex for this contrast, even at a lower threshold.

Clusters with significant activation in other contrasts for the Hand, Ghost, and Text conditions during the execution phase are included in tables in the S1 File. Reaction times are described in a previous publication [20].

## Discussion

Our aim in this study was to directly contrast two hypotheses concerning the brain areas involved in (delayed) imitation. Heyes' Grist-Mills hypothesis claims that performing an imitative action uses the same brain mechanisms as performing a matched non-imitative action because the mechanisms of social learning "are the same associative mechanisms that encode information [. . .] when it is derived, not from observing the behaviour of others (social learning), but from direct interaction with the inanimate world (asocial learning)" ([14], p. 2183). In contrast, mirror neuron theories suggest that there are dedicated brain areas and systems

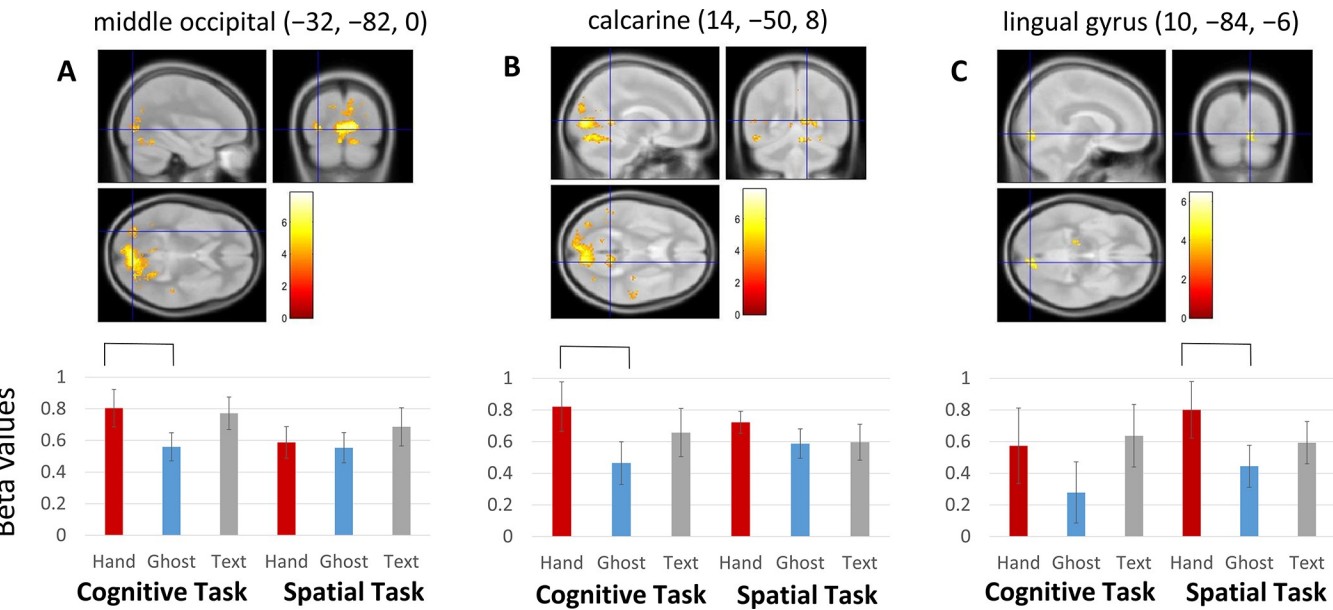

**Fig 2.** Regions showing greater activity when executing a sequence following a Hand demonstration (red bars) compared to a Ghost demonstration (blue bars) in the cognitive task (panels A and B) and in the spatial task (C). Brackets above the bars indicate which contrasts are significantly different and the focus of the brain plot above. Error bars indicate 95% confidence intervals calculated using within-subjects standard error [29].

**Table 1. Cognitive task, Hand > Ghost: Brain regions, FWE-corrected p values, sizes of clusters (k), T values, and MNI coordinates of clusters showing significant activation following different demonstration types.**

| Brain region | p(FWE) | k | T | MNI coordinate | | |
|---|---|---|---|---|---|---|
| | | | | x | y | z |
| R calcarine | <0.001 | 3629 | 7.87 | 14 | −50 | 8 |
| | | | 7.56 | 8 | −84 | 2 |
| | | | 7.45 | 10 | −76 | 4 |
| L middle occipital | <0.001 | 198 | 5.57 | −32 | −82 | 0 |
| | | | 4.94 | −38 | −78 | 8 |
| | | | 4.62 | −34 | −74 | 2 |
| R precuneus | 0.01 | 74 | 5.32 | 2 | −46 | 48 |
| L precuneus | | | 4.37 | 0 | −56 | 48 |
| L middle cingulate | | | 3.68 | −4 | −40 | 52 |
| R parahippocampus | 0.02 | 65 | 5.03 | 24 | −22 | −22 |
| R fusiform | | | 4.23 | 22 | −30 | −18 |
| L amygdala | 0.04 | 55 | 4.89 | −22 | −6 | −18 |
| L hippocampus | | | 4.27 | −28 | −12 | −12 |

for imitation. Although parts of the MNS in premotor or parietal cortex have been linked to the learning of distinct responses *during demonstration* [20], the results of the present study did not show any evidence that they play a specific role in *executing* imitated responses learned vicariously from an agent. Instead, consistent with a growing body of research [18, 19], we found significant activation in regions outside the classic mirror system. We discuss the implications of these results for theories of imitation.

## The lack of imitation-specific responses in mirror regions

We did not find imitation-specific responses in MNS regions. This contrasts with previous studies which have reported activation in MNS areas during imitation tasks [4, 5, 17, 30]. However, such immediate imitation studies used paradigms in which action observation and execution occurred simultaneously; this generally meant that visual input during imitation conditions was different from that during non-imitation conditions. Consequently, it is difficult to interpret the resulting brain activity as being due strictly to imitation itself.

A stronger test of these hypotheses is to examine delayed imitation, but this has rarely been done. While one previous study using a delayed imitation task reported activity in the MNS [2], it did not directly contrast the imitative versus non-imitative (freely chosen) actions. Another experiment testing delayed hand posture matching [19] did not find strong

**Table 2. Spatial task, Hand > Ghost: Brain regions, FWE-corrected p values, sizes of clusters (k), T values, and MNI coordinates of clusters showing significant activation following different demonstration types.**

| Brain region | p(FWE) | k | T | MNI coordinate | | |
|---|---|---|---|---|---|---|
| | | | | x | y | z |
| R lingual | <0.001 | 202 | 6.46 | 10 | −84 | −6 |
| | | | 5.70 | 16 | −78 | −8 |
| | | | 4.17 | 14 | −84 | −14 |
| Vermis | 0.032 | 69 | 4.78 | 2 | −34 | −20 |
| L cerebellum | | | 4.13 | −4 | −36 | −26 |
| L cerebellum | | | 4.11 | −4 | −30 | −10 |

engagement of mirror systems. Consistent with this latter study, we also found no MNS activity during the execution portion of the task.

The lack of mirror system activation in the execution phase of our study cannot be attributed to a failure to engage these brain regions at all. In the demonstration phase of both the cognitive and spatial tasks with a visible hand, there was activity in MNS areas: in the cognitive task, the Hand > Text contrast revealed an activation difference in right inferior frontal gyrus (IFG). In the spatial task, the Hand > Text contrast showed a difference in left parietal cortex [20]. However, these differences occurred during the demonstration phase only. As such, MNS activity during imitation conditions appears to be restricted to action observation alone, not both action observation *and* execution, as the MNS hypothesis would predict.

As with some previous neuroimaging studies, the sample size was not large. Yet in both phases of the experiment (demonstration and execution), power was sufficient to detect the differential activation of brain areas in the three conditions. The interpretation of null findings in event-related fMRI is challenging; it could be that MNS areas were active during execution but to a lesser degree than occipital areas. However, even with a lower threshold, we did not find MNS areas to be active during action execution. We maintain that the finding of no differential activation in the MNS during imitative action execution cannot be attributed to insufficient power.

## The engagement of occipitotemporal cortex by imitation

In contrast to the lack of activation of MNS regions, we found robust engagement of occipital and occipitotemporal cortex in our tasks. Specifically, when participants executed the cognitive task following a Hand action demonstration (imitation), calcarine sulcus and middle occipital regions were activated. When participants executed the spatial task following a Hand action demonstration, lingual gyrus was activated. These results are again in line with the finding from Makuuchi [19] showing that occipital regions can be engaged when performing an imitative action.

There is evidence from a variety of sources that regions within medial and lateral occipital cortex have an important role in action execution. A careful study from Astafiev and colleagues showed that the extrastriate body area (EBA), in addition to its role in passive observation of bodies and body parts, also responds to self-generated movement [31]. When contrasted under conditions of covert attention, saccades, and pointing actions, EBA showed significantly greater activation for pointing. This demonstrates movement-related modulation of the EBA which cannot be attributed to simple attentional or visual factors of the target stimulus. In addition, a multivoxel pattern analysis study examined the coding of action observation and action performance across the brain [32] and found evidence of cross-modal action representations in occipitotemporal cortex (as well as parietal cortex). These previous results [31, 32] are in agreement with results from a meta-analysis [3] showing that occipitotemporal cortex, along with the MNS, is involved in imitation. Our data corroborate these prior studies' results. Specifically, our study shows that regions of occipitotemporal cortex have a specific role in the execution of imitative responses following a Hand demonstration—neural activity not seen for identical responses following a non-social Ghost demonstration.

If we seek to understand neural mechanisms of imitative learning, we need to include occipital regions too. EBA and the other parts of occipital cortex highlighted here respond to actions of self and other and to imitation. In particular, the finding that the only brain regions in the cortex which are engaged *specifically* for imitative actions and not for matched non-imitative actions are located in the medial and lateral occipital cortex suggests that the previous focus on the roles of inferior parietal lobule (IPL) and IFG in imitation has limited our understanding. Further research into the role of occipital cortex in imitation could determine the factors modulating its involvement.

Could the occipital activity in the execution phase be due to "spillover" from the demonstration phase due to the timing of the hemodynamic response function (HRF)? We think this is unlikely for the following reason. For the cognitive task, the occipital areas activated during the demonstration phase for the Hand > Ghost contrast are bilateral occipital gyrus and temporal gyrus (clusters centered on coordinates −54 −74 10 and 56 −70 0). The activated clusters from the execution phase are right calcarine on the medial brain surface (cluster centered on coordinates 14 −50 8) and left middle occipital gyrus (coordinates −32 −82 0), posterior to the cluster from the demonstration phase. For the spatial task, the corresponding areas for the demonstration phase are left occipital gyrus and temporal gyrus (coordinates −50 −70 6) and right occipital gyrus (26 −100 −6), while the execution phase cluster is located in the right lingual gyrus (10 −84 −6), also on the medial brain surface. If the areas activated during the execution phase were simply a result of completion of the HRF initiated due to the demonstration phase stimuli, we would expect them to be the same brain area, rather than different areas.

## Grist and mills

Our findings have important implications for the Grist-Mills hypothesis. Heyes claims that social and non-social learning rely on the same basic learning mechanisms (the mill), but that input mechanisms—motivational, perceptual, and attentional processes—may be biased toward information (grist) from social sources [14, 15]. If there is essentially one "mill" that performs all types of learning (whether socially mediated or not), then there should not be detectable brain differences between executing an imitative and a non-imitative action, especially in studies like this one where imitation is delayed and the actions are precisely matched for motor complexity and visual feedback. In fact, our results show that despite the fact that the execution phases across conditions were matched, executing a sequence following a Hand demonstration produces different brain activation patterns from executing a sequence following a Ghost or Text demonstration. That is, social learning involves different neural correlates from non-social learning. This outcome is inconsistent with the core prediction of the Grist-Mills hypothesis, as it strongly indicates that different mechanisms (mills) are involved in learning from social (Hand) and non-social (Ghost) sources.

These activation differences could arise for several reasons. In a recent study of action observation and simultaneous action, participants were instructed to imitate deliberately or to perform a certain action which happened to be imitative [33]. Mid-occipital regions were specifically engaged for intentional imitation, and also showed more activity when participants' actions matched those observed regardless of instructions. These findings were interpreted in terms of differences in motor planning and attention, with more attention paid when participants were instructed to imitate. However, the results from Astafiev and colleagues [31] suggest that attention alone does not account for different patterns of activity in lateral occipital cortex during motor tasks, and in our study the visual inputs were matched during task execution, which suggests that visual attention should also be matched.

Another possibility might draw on Meltzoff's original active intermodal mapping (AIM) model, where an "equivalency detector" responds to the correspondence between, for example, the action observed on the face of an adult and the action performed by an infant [34]. A related cognitive framework assumes that social and asocial learning conditions produce distinct cognitive models (or simulations) against which performance during action execution is compared [35]. We might think of occipital and occipitotemporal regions implicated in imitation execution as areas which use the visual input of the demonstrator's hand actions to generate a model based on what is known about the task (e.g., the content of the demonstration). The model makes predictions about the serial hand actions which the participant will perform,

testing these predictions against the motor and proprioceptive feedback from the actual—ongoing—hand actions (for a related model of predictive coding in the MNS, see [36]). However, the differing patterns of activation between the imitation of item-specific sequences in the cognitive task and location-specific sequences in the spatial task suggest that there is no single brain area in which equivalencies are detected or model-based predictions are compared with feedback from either past or present experiences. Rather, multiple areas may perform distinct computations depending on task demands [12, 37].

## Conclusion

Here, we explored brain responses during imitative and matched non-imitative action execution, contrasting the MNS hypothesis and the Heyes Grist-Mills hypothesis [14–16]. While the MNS appears to be involved in observational learning (i.e., action observation), we found no evidence that the classic mirror neuron regions of IPL and IFG respond selectively during the execution of imitative responses. Instead, medial and lateral occipital cortex appear to be specifically engaged during the execution of imitative responses. This is consistent with previous results and implies that the study of the neural basis of imitation needs to look beyond the mirror neuron system that presently includes only parietal and inferior frontal regions. The results also failed to support the Grist-Mills hypothesis and the idea that there is nothing special about how the brain learns from and imitates others. Future research is necessary to understand the information processing that is taking place in these brain regions.

## Supporting information

**S1 File.**
(DOCX)

## Acknowledgments

We thank the Sir Peter Mansfield Magnetic Resonance Imaging Centre at the University of Nottingham for help with fMRI.

## Author Contributions

**Conceptualization:** Francys Subiaul, Antonia F. de C. Hamilton.

**Data curation:** Elizabeth Renner, Yishan Xie, Antonia F. de C. Hamilton.

**Formal analysis:** Elizabeth Renner, Yishan Xie.

**Funding acquisition:** Elizabeth Renner, Francys Subiaul, Antonia F. de C. Hamilton.

**Investigation:** Elizabeth Renner.

**Methodology:** Francys Subiaul, Antonia F. de C. Hamilton.

**Software:** Antonia F. de C. Hamilton.

**Supervision:** Antonia F. de C. Hamilton.

**Visualization:** Elizabeth Renner, Antonia F. de C. Hamilton.

**Writing – original draft:** Elizabeth Renner, Antonia F. de C. Hamilton.

**Writing – review & editing:** Elizabeth Renner, Yishan Xie, Francys Subiaul, Antonia F. de C. Hamilton.

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
