## [Decision Letter · Decision Letter 0]

24 May 2023

PONE-D-23-06849Are there dedicated neural mechanisms for imitation? A study of grist and millsPLOS ONE

Dear Dr. Renner,

Thank you very much for submitting your manuscript "Are there dedicated neural mechanisms for imitation? A study of grist and mills" for review and consideration for publication in PLOS ONE.I have now received comments from two external reviewers. Although both found merit in your paper, they also both identified a number of issues. For your guidance, reviewers' comments are appended below.Therefore, I invite you to submit a revision together with a cover letter explaining how you have responded to the reviewers’ comments.

We look forward to receiving your revised manuscript.

Kind regards,

Cédric A. Bouquet

Academic Editor

PLOS ONE

Journal Requirements:

Reviewers' comments:

Reviewer's Responses to Questions

**Comments to the Author**

1. Is the manuscript technically sound, and do the data support the conclusions?

Reviewer #1: Yes

Reviewer #2: Partly

2. Has the statistical analysis been performed appropriately and rigorously? 

Reviewer #1: Yes

Reviewer #2: Yes

3. Have the authors made all data underlying the findings in their manuscript fully available?

Reviewer #1: Yes

Reviewer #2: Yes

4. Is the manuscript presented in an intelligible fashion and written in standard English?

Reviewer #1: Yes

Reviewer #2: Yes

5. Review Comments to the Author

Reviewer #1: This paper looks at brain activity during action execution in a delayed imitation task to compare two hypotheses of imitation: that it relies on dedicated brain mechanisms (MNS hypothesis) and that it relies on more general sensorimotor processing (Grist-Mills hypothesis). The results show imitative action execution elicits more occipital brain activity than non-imitative action execution. However, no differences are found in MNS areas.

I think this is an interesting, well-designed, and clearly-written study. I only have a couple of minor comments that I hope may help to further improve the paper:

1. It’s not entirely clear to me why the MNS hypothesis would necessarily predict more brain activity in MNS areas during execution of delayed imitation. Is it because the MNS hypothesis assumes that these areas are not only involved in executing the action, but also in matching the observed action to a motor command? If so, predicting more activity for imitation in the execution phase implies that this matching process has not yet been done by then. Could the authors unpack their prediction a bit more and explain exactly why the MNS hypothesis predicts stronger brain activity during execution?

2. Similarly, it’s not entirely clear to me why the Grist-Mills hypothesis predicts no differences in brain activity. Couldn’t differences in brain activity also reflect differences in input? For example, as the authors suggest in the discussion, it could be that “hand input” and “ghost input” is maintained differently between observation and execution. If true, the mechanisms transforming the visual input to a motor response would still be the same.

3. Why are the execution events modeled as 6s events and not as events with duration equal to the total execution time? If a fixed duration is used, couldn’t execution time influence brain activity?

4. The authors write that the absence of activity differences in MNS areas can’t be ascribed to low power because they find differences in other areas. I don’t think this claim is justified, as power of course depends on the effect size. It could also be that there are differences in MNS areas but that these differences are smaller than those in occipital areas.

Reviewer #2: This paper revisits the question of which brain regions are involved in imitation. Although several fMRI studies addressed this timely question, most suffer from inadequate controls, as discussed in the present paper. Here, the authors contrasted two hypotheses of imitation: the well-known mirror-neuron system hypothesis, implying parieto-frontal regions, and a hypothesis that proposes no differences between social and non-social processes, implying the involvement of the same brain regions in imitation as in non-social performance. They re-analyzed data from their previously published fMRI experiment and their data do not support either hypothesis. The most striking observation was the absence of mirror neuron system imitation-related activations, but null results are difficult to interpret, especially in event-related fMRI. My main concern relates to the interpretation of the occipital activations which the authors attribute to imitation. I believe that more control analyses (and perhaps experiments) are required to make such a conclusion

Main comments.

1. I was wondering whether the occipital activations they observed during the execution phase are not because of a spillover from the activations to the preceding different demonstration conditions (BOLD is a very slow response). Did those activations depend on the time interval between the demonstration and execution phase, being stronger with shorter intervals? If so, then this would support the trivial explanation of spillover. This should be addressed.

2. The “visual” activations appear to include V1 and I find it difficult to believe that V1 would be involved in imitation, except for differences in visual processing between conditions. The same may hold for the more anterior occipital activations (EBA?). Related to this issue, did the viewing (fixation/saccade) patterns during the execution phase differ among the different conditions? Also, were the reaction times the same in the different conditions? Different reaction times relate to differences in exposure to the stimuli and thus differences in occipital activations.

3. The authors contrasted the hand and ghost conditions, ignoring the text condition. However, Figure 2 suggests similar activations between the hand and text condition and also between the text and ghost condition in the occipital areas. They should consider the text condition activations also in the description of the Results and the interpretation.

4. The text condition activations, in comparison to those of the hand condition, make me wonder to what extent the activations in the occipital areas are related to imitation and do not result from confounding, ill-controlled effects of the preceding demonstration phases or during the execution phase (see main points 1 and 2 above). As long as the authors cannot make a convincing case to the contrary, the evidence of occipital (and EBA) being involved in imitation is weak (if not absent), and that should be discussed.

5. The absence of mirror neuron system activation is the most interesting finding of this experiment. However, as I mentioned above, null results in event-related fMRI are difficult to interpret, and that should be acknowledged.

6. PLOS authors have the option to publish the peer review history of their article (what does this mean?). If published, this will include your full peer review and any attached files.

Reviewer #1: No

Reviewer #2: No

---

## [Author Response · Author response to Decision Letter 0]

1 Aug 2023

Please see the uploaded response to reviews document.

---

## [Decision Letter · Decision Letter 1]

5 Sep 2023

Are there dedicated neural mechanisms for imitation? A study of grist and mills

PONE-D-23-06849R1

Dear Dr. Renner,

I am pleased to inform you that your manuscript has been judged scientifically suitable for publication and will be formally accepted for publication once it meets all outstanding technical requirements.

As you will see below, both reviewers are happy with the revision and feel that all their concerns have been addressed.

Thank you again for this very interesting paper.

Kind regards,

Cédric A. Bouquet

Academic Editor

PLOS ONE

Additional Editor Comments (optional):

Reviewers' comments:

Reviewer's Responses to Questions

**Comments to the Author**

1. If the authors have adequately addressed your comments raised in a previous round of review and you feel that this manuscript is now acceptable for publication, you may indicate that here to bypass the “Comments to the Author” section, enter your conflict of interest statement in the “Confidential to Editor” section, and submit your "Accept" recommendation.

Reviewer #1: All comments have been addressed

Reviewer #2: All comments have been addressed

2. Is the manuscript technically sound, and do the data support the conclusions?

Reviewer #1: Yes

Reviewer #2: Yes

3. Has the statistical analysis been performed appropriately and rigorously? 

Reviewer #1: Yes

Reviewer #2: Yes

4. Have the authors made all data underlying the findings in their manuscript fully available?

Reviewer #1: Yes

Reviewer #2: Yes

5. Is the manuscript presented in an intelligible fashion and written in standard English?

Reviewer #1: Yes

Reviewer #2: Yes

6. Review Comments to the Author

Reviewer #1: (No Response)

Reviewer #2: I am happy to see that the authors responded to all my comments and I am satisfied with their revisions.

7. PLOS authors have the option to publish the peer review history of their article (what does this mean?). If published, this will include your full peer review and any attached files.

Reviewer #1: No

Reviewer #2: No

---

## [Editor Report · Acceptance letter]

18 Sep 2023

PONE-D-23-06849R1 

Are there dedicated neural mechanisms for imitation? A study of grist and mills 

Dear Dr. Renner:

I'm pleased to inform you that your manuscript has been deemed suitable for publication in PLOS ONE. Congratulations! Your manuscript is now with our production department. 

Kind regards, 

on behalf of

Dr. Cédric A. Bouquet 

Academic Editor

PLOS ONE